# Effects of Prolonged Fasting during Inpatient Multimodal Treatment on Pain and Functional Parameters in Knee and Hip Osteoarthritis: A Prospective Exploratory Observational Study

**DOI:** 10.3390/nu15122695

**Published:** 2023-06-09

**Authors:** Daniela A. Koppold, Farid I. Kandil, Oliver Güttler, Anna Müller, Nico Steckhan, Sara Meiß, Carolin Breinlinger, Esther Nelle, Anika M. Hartmann, Michael Jeitler, Etienne Hanslian, Jan Moritz Fischer, Andreas Michalsen, Christian S. Kessler

**Affiliations:** 1Institute of Social Medicine, Epidemiology and Health Economics, Charité—Universitätsmedizin Berlin Corporate Member of Freie Universität Berlin and Humboldt-Universität zu Berlin, 10117 Berlin, Germany; farid-ihab.kandil@charite.de (F.I.K.); oliver.guettler@charite.de (O.G.); anna.mueller@charite.de (A.M.); nico.steckhan@hpi.de (N.S.); sara.meissheit@gmail.com (S.M.); carolin.breinlinger@nu.uni-giessen.de (C.B.); esther.nelle@charite.de (E.N.); anika.hartmann@charite.de (A.M.H.); michael.jeitler@charite.de (M.J.); etienne.hanslian@charite.de (E.H.); jan-moritz.fischer@charite.de (J.M.F.); andreas.michalsen@immanuelalbertinen.de (A.M.); christian.kessler@charite.de (C.S.K.); 2Department of Internal Medicine and Nature-Based Therapies, Immanuel Hospital Berlin, 14109 Berlin, Germany; 3Department of Pediatrics, Division of Oncology and Hematology, Charité—Universitätsmedizin Berlin, Corporate Member of Freie Universität Berlin and Humboldt-Universität zu Berlin, 10117 Berlin, Germany; 4State Institute of Forensic Medicine Berlin, 10559 Berlin, Germany; 5Connected Healthcare, Hasso Plattner Institute, University of Potsdam, 14482 Potsdam, Germany; 6Department of Dermatology, Venereology and Allergology, Charité—Universitätsmedizin Berlin, Corporate Member of Freie Universität Berlin and Humboldt-Universität zu Berlin, 10117 Berlin, Germany

**Keywords:** fasting, caloric restriction, osteoarthritis, dietary intervention, fasting-mimicking diet, integrative medicine, complementary medicine, traditional European medicine, nutrition, multimodal integrative treatment

## Abstract

Preliminary clinical data suggest that pain reduction through fasting may be effective for different diagnoses. This uncontrolled observational clinical study examined the effects of prolonged modified fasting on pain and functional parameters in hip and knee osteoarthritis. Patients admitted to the inpatient department of Internal Medicine and Nature-based Therapies of the Immanuel Hospital Berlin between February 2018 and December 2020 answered questionnaires at the beginning and end of inpatient treatment, as well as at 3, 6, and 12 months after discharge. Additionally, selected blood and anthropometric parameters, as well as subjective pain ratings, were routinely assessed during the inpatient stay. Fasting was the only common intervention for all patients, being performed as part of a multimodal integrative treatment program, with a daily caloric intake of <600 kcal for 7.7 ± 1.7 days. N = 125 consecutive patients were included. The results revealed an amelioration of overall symptomatology (WOMAC Index score: −14.8 ± 13.31; *p* < 0.001; d = 0.78) and pain alleviation (NRS Pain: −2.7 ± 1.98, *p* < 0.001, d = 1.48). Pain medication was reduced, stopped, or replaced by herbal remedies in 36% of patients. Improvements were also observed in secondary outcome parameters, including increased quality of life (WHO-5: +4.5 ± 4.94, *p* < 0.001, d = 0.94), reduced anxiety (HADS-A: −2.1 ± 2.91, *p* < 0001, d = 0.55) and depression (HADS-D: −2.3 ± 3.01, *p* < 0.001, d = 0.65), and decreases in body weight (−3.6 kg ± 1.65, *p* < 0.001, d = 0.21) and blood pressure (systolic: −6.2 ± 15.93, *p* < 0.001, d = 0.43; diastolic: −3.7 ± 10.55, *p* < 0.001, d = 0.43). The results suggest that patients with osteoarthritis of the lower extremities may benefit from prolonged fasting as part of a multimodal integrative treatment to improve quality of life, pain, and disease-specific functional parameters. Confirmatory randomized controlled trials are warranted to further investigate these hypotheses.

## 1. Introduction

Osteoarthritis (OA) is the most common joint disorder and a prominent cause of disability worldwide. It involves the inflammation and destruction of cartilaginous and adjacent bone structures, leading to pain, stiffness, and disability [1,2,3]. Morbidity has been estimated at about 17.9% in German adults, its prevalence increasing with age, and it is more common among the female gender [1]. Impairments to overall mobility are most pronounced in lower limb OA; approximately half of OA patients have at least one knee joint affected, and a quarter have at least one hip joint affected [1]. OA’s symptoms, ranging from pain during physical activity to ankylosis, can include pain during rest or sleep, stiffness, and gait unsteadiness. OA also seems to be associated with a higher incidence of mental health disorders, such as depression and suicidality, as well as higher cardiovascular risk [3]. Hence, it is obvious that OA consumes a substantial amount of healthcare resources, primarily due to the costs of joint replacement surgery [3].

Nutritional factors have been found to influence the prevention and management of knee and hip OA. The mechanisms which have been discussed include weight loss, reducing inflammation, and promoting antioxidation [4]. This has led to plant-based diets and the Mediterranean diet being suggested for patients suffering from OA of the lower limbs [4].

Fasting has been shown to activate, enhance, and accelerate analogous mechanisms, inducing weight loss, anti-inflammation, cell-repair, stem cell production, and antioxidation [5]. It also seems to improve mood and reduce pain perception due to serotonin enhancement [6]. Additionally, it can positively influence cardiometabolic risk factors by increasing insulin sensitivity, reducing mammalian target of rapamycin (mTOR) activity, and reducing unfavorable blood lipids [7,8]. The gut microbiome also changes during fasting, and it has been recently suggested by human and animal data that the microbiome is associated with OA disease progression [9]. Moreover, fasting has been reported to support long-term dietary and lifestyle changes by enhancing self-efficacy, opening a window of opportunity for creating new habits [10] and sensitizing taste, especially to salt and sugar.

It seems reasonable to infer that fasting could possibly support conventional OA care, as experimental data indicate that it influences short and long-term pathomechanisms fueling OA progression. In one small uncontrolled study, *n* = 8 hip and *n* = 12 knee OA patients fasted for one week, with a daily intake of 300 kcal. The results showed pain reduction; improved articular function; and improvements in weight, body mass index (BMI), and waist circumference [11]. Furthermore, there is evidence from clinical studies and a systematic review supporting the efficacy of fasting for treating symptoms of rheumatoid arthritis [12,13,14,15].

In Germany, inpatient fasting in hospitals is usually offered as the main aspect of a multimodal individualized treatment plan based on traditional, nature-based European medical concepts in some specialized departments [7,16]. For patients with OA, this can include physiotherapy, moderate exercise, dietary counseling, cold or warm local applications, herbal medicines, and other methods. This renders it impossible to observe the effect of fasting only, as patients do not follow only one therapy at a time, as would be necessary for an effective randomized controlled trial. Instead, it provides an excellent opportunity to examine the real-world implications of therapeutic fasting, as patients usually combine therapeutic options in clinical settings.

The aim of this observational study was to observe the effects of a therapeutic prolonged fast offered to patients suffering from knee and/or hip OA during multimodal hospital treatment, with a focus on nature-based traditional European medicine (TEM) and complementary and integrative medicine (CIM).

## 2. Materials and Methods

### 2.1. Study Design

This trial was designed as an explorative, single-arm, prospective, single-center, open-label, observational study. The study protocol was approved by the institutional review board of Charité Universitätsmedizin Berlin (Charitéplatz 1, 10117 Berlin) in October 2015 (ID: EA4/005/17). It was registered with ClinicalTrials.gov (ClinicalTrials ID: NCT03785197) (accessed on 24 April 2023) and was conducted according to the standards of the Declaration of Helsinki. Written informed consent was obtained from all participants prior to their entry into the study.

### 2.2. Setting

Participants were recruited between February 2018 and December 2020 at the Immanuel Hospital Berlin. This is a hospital with 195 beds and approximately 5000 inpatient admissions per year, including orthopedic, surgical, rheumatological, and osteological wards. There is also a ward for internal medicine and nature-based therapies (IMNT), consisting of 60 beds and a day care clinic using nature-based (NB) and mind–body medicine approaches. Most patients are admitted by German statutory health insurance coverage, and a few come as privately insured patients or self-payers.

The Department of IMNT at the Immanuel Hospital Berlin is one of the leading institutions in terms of the application of NB and traditional medicinal approaches, including TEM [17,18,19,20,21,22,23,24,25,26,27,28,29]. This department is especially experienced in applying prolonged fasting as a therapeutic intervention for diverse disease entities [6,16,30,31,32,33,34,35,36,37,38,39,40,41,42,43,44,45,46,47,48,49,50,51].

We decided to focus on the four most prevalent diagnoses for inpatient treatment in 2017, which included osteoarthritis (knee and hip ICD codes: M17.9 and M16.9, respectively). The other diagnoses were rheumatoid arthritis (M06.9), fibromyalgia syndrome (M79.7), and type 2 diabetes mellitus (E11.61), the results for which will be reported elsewhere.

To be admitted to the IMNT ward for OA treatment, patients needed to have been prediagnosed by specialists such as orthopedic surgeons or rheumatologists.

After admission to the inpatient department of IMNT and the first medical consultation, the main diagnosis for the inpatient treatment and an individual, multimodal treatment plan, including a dietary regimen, were determined. Based on these, study personnel screened patients for eligibility. If eligible, patients were informed in detail about the option to participate in the observational study. For details on eligibility criteria, please refer to “Section 2.4 Participants”.

Study visits at the beginning and end of the inpatient treatment included validated questionnaires on patient-reported outcomes. The questionnaires were handed to the patients on a tablet in electronic form on the first or second day, as well as on one of the last two days, of the inpatient treatment. The laboratory tests included in the standard procedures of the inpatient ward at admission and discharge were also assessed for study purposes. Data on the use of pain medication, blood pressure, body weight, the side effects of fasting, and their management were extracted from the documentation made by nurses and doctors in the patient record after patients were discharged.

During their hospital stay, patients usually received visits from the ward’s physicians four to five times a week, while the nursing staff had daily contact with each patient. On the first day of the inpatient stay, a blood test was routinely performed. These blood tests usually included a complete blood count without differential, blood glucose, blood lipids, electrolytes, and standard parameters of kidney and liver function; if there were findings that needed to be controlled, another blood test was prescribed by the responsible physician and performed towards the end of the inpatient stay. This routine was established before the commencement of this study, and remained unchanged throughout the entire study period.

Follow-ups after three, six, and twelve months were conducted through questionnaires only, either electronically (if e-mail addresses were provided by the participants) or by mail.

### 2.3. Interventions

When fasting is prescribed by a physician as part of routine inpatient care, the first complete day of the hospital stay is a preparatory day, with a light plant-based diet. The procedure is based on the consensus guidelines for fasting therapy [16]. On this day, the patient eats a calorie-reduced light diet of approximately 1200 kcal. This is accompanied by a TEM bowel cleansing procedure, which is induced by taking laxatives, such as Glauber’s salt, or an enema. On the second day, fasting starts. It is usually scheduled for at least five consecutive days and a maximum of twelve days, with the length depending on the individual constitution of the patient and the regulation of the inpatient stay according to diagnosis, disease severity, and ICD-10 code. During fasting, only natural juices, unsweetened herbal teas, and water are consumed. The daily caloric intake is between 200 and 300 calories. During the SARS-CoV-2-pandemic, starting from April 2020 and lasting until the end of the study in December 2020, the prescribed fasting regimen was modified to a fasting-mimicking diet (FMD) due to uncertainties about the role of fasting in the immunological response to SARS-CoV-2. Daily caloric intake was raised to a maximum of 600 kcal, including solid foods, such as porridge, potatoes, steamed vegetables, and vegetable soups. The foods did not contain sugar or sweeteners, and the salt content was reduced.

Inpatient fasting treatment is embedded in a set of other therapeutic interventions which are prescribed by a doctor for each patient individually. The multimodal CIM/TEM treatment program is a defined and established concept, commonly reimbursed by German statutory health insurance companies. It consists of at least 120 min of non-pharmacological treatment modalities and lifestyle counseling per day and is tailored according to the individual patient’s diagnosis and needs. These interventions comprise nutritional counseling, exercise, physiotherapy, thermal treatments, mind–body medicine elements, and other aspects of traditional medicine. These treatments are a recommended part of traditional fasting, as published earlier [16].

Fasting duration is mainly determined by the constitution of the patient, including BMI, body composition, the dynamics of weight loss during fasting and vitality, as well as subjective criteria such as well-being and feelings of hunger or exhaustion. The length of the fast is thus decided between the therapeutic team and the patient during the visits.

After the fasting days are complete, the fast is broken on the last day by eating an apple. The following one to three days, depending on the length of the fast, solid foods are gradually reintroduced in the form of a light plant-based diet. During this period, medication that was paused or adapted during fasting to ensure a safe fast, is gradually reintroduced where necessary. After the fasting cure, usually a normocaloric plant-based diet is last recommended as a follow-up diet.

### 2.4. Participants

All patients with OA as their main diagnosis who received inpatient treatment including therapeutic fasting at the Department for IMNT at the Immanuel Hospital Berlin between February 2018 and December 2020 were screened for eligibility.

Inclusion criteria included age between 18 and 85 years and written informed consent provided during the first 24 h of the inpatient treatment.

Patients with symptomatic gallstones, a history of gout or eating disorders, cachexia or sarcopenia, acute psychosis, severe psychiatric pathologies, and severe metabolic conditions such as liver or kidney dysfunction insufficiency were excluded, as these conditions present contraindications for fasting [16].

Further exclusion criteria were language barriers, dementia or other major cognitive impairments, pregnancy or lactation, and participation in any other study.

### 2.5. Variables

The main outcome was assessed by the Western Ontario and McMaster Universities Osteoarthritis Index (WOMAC, Version 3.1), a validated questionnaire specific to osteoarthritis of the knee and the hip. We used the WOMAC Index in the 5-point Likert-scale version that gives a global score in a range between 0 and 96 points. Clinically relevant changes/minimal clinically important differences (MCID) for the WOMAC Index are 10 points for the newer 0–240 VAS/NRS scale [52] and, thus, 4 points for the 0–96 scale used herein. Secondary outcomes included the Hospital Anxiety and Depression Scale (HADS), the Mindful Attention Awareness Scale (MAAS), and the WHO Quality of Life questionnaire (WHO-5). We also collected data regarding acute pain on a numerical rating scale (NRS), as documented in the patient record. A clinically relevant change for pain on the NRS was defined as a reduction of one point, while a NRS change score of −2.0 was associated with the concept of “much better” improvement [53]. Furthermore, body weight, blood pressure, and medication were extracted from the patient record, as well as triglycerides, total cholesterol, LDL, and HDL.

### 2.6. Data Collection/Measurement

Questionnaires were filled in digitally with tablets during the inpatient stay and in digital or analog form at follow-ups, depending on the patient’s preference. Pain was documented on the NRS scale, and the side-effects of fasting were assessed several times during the inpatient stay. For the calculation of pain scores on the NRS, we selected the first and last documented scores in the patient record, as for most patients, these were first and last days of their inpatient stays. Blood samples that were collected under routine care conditions on the first day and, if deemed necessary by the responsible physician, on one of the last two days of the inpatient stay as well, were used for the study’s purposes. The fasting blood samples were taken by hospital physicians between 7.30 and 8.15 AM each day, before breakfast.

### 2.7. Bias

All limitations of observational studies apply here, among those being a missing control group and lack of randomization, as well as the inevitable impossibility of blinding patients in nutritional interventions. Fasting, especially, cannot be blinded per se, neither for the patient nor for the hospital personnel.

The study personnel were only involved in the recruitment of the participants and in ensuring that they filled out the questionnaires; they were not involved in any other aspect of the inpatient hospital stay. As such, the study personnel had no influence on the length of the fasting period, any adjustments to therapeutic modalities, or any other direct or indirect influence on the patients’ courses of therapy.

To detect any reporting bias connected to subjective improvement or deterioration of symptoms during follow-ups, we controlled for strong positive or negative responses to fasting.

### 2.8. Study Size

Based on the data obtained in previous years, it was estimated that during the 3 years of the study, *n* = 150 patients with OA, of which *n* = 125 would agree to participate in this study, would be admitted to the ward and would be prescribed therapeutic fasting as part of their inpatient treatment. In this setting of an exploratory pre-post-comparison using the *t*-test and standard parameters of alpha = 0.05 and beta = 0.20 (corresponding to a power of 80%), a number of *n* = 125 patients was sufficient to detect all large, medium, and small effects with a minimal effect size of Cohen’s d ≥ 0.23. No interim analyses were planned or performed.

### 2.9. Statistical Methods

In this explorative, observational, single-arm study, *t*-tests were used to compare participant’s scores and vital parameters between baseline (V0) and subsequent visits (V1 = at discharge, V2 = 3 months after baseline, V3 = 6 months after baseline, and V4 = 12 months after baseline) by means of unadjusted *t*-tests. To estimate the effect size of the interaction, data for the primary endpoint (WOMAC and its subscales) were additionally analyzed for the subgroups of knee and hip OA by ANOVAs using the group affiliation (knee/hip) as a second factor next to the visits. As is usual in exploratory studies, no correction for multiple testing (alpha adjustment) was applied. All analyses were based on the set of complete cases available for the individual questionnaire or parameter.

To determine whether there was a reporting bias connected to subjective improvement or deterioration of symptoms in the follow-ups, patients were subdivided according to their WOMAC score (primary endpoint) improvements at V1 into high, medium, and low gainers. For subsequent follow-ups, we cross-checked whether any subgroups were under- or overrepresented.

## 3. Results

In this longitudinal and uncontrolled observational study, *n* = 125 hospitalized patients (*n* = 107 females and *n* = 18 males) with knee or hip OA undergoing inpatient prolonged medical fasting between February 2018 and December 2020 were recruited. For baseline characteristics, please refer to Table 1. In *n* = 97 (78%) participants, OA mainly affected the knee joint(s); in *n* = 28 (22%), the hip joints. Patients were mostly between *n* = 50 and *n* = 65 years of age, with an average of 61.3 (±10.2) years. The majority (*n* = 108, 86.4%) reported a moderate to strong subjective sense of ill physical health during the two weeks preceding inpatient treatment, while *n* = 55 (44%) stated that there had been a moderate to strong affection of their psychological health in the same time frame. While most patients (*n* = 86, 68%) were familiar with the concept of integrative medicine, for *n* = 83 (66.4%), it was the first hospital admission of this kind, and *n* = 56 (44.8%) did not have previous experience with therapeutic fasting. However, the expectations of efficacy were moderately high (6.4 ± 2.0) on an NRS (0–10) scale (Table 1).

During the stay, *n* = 18 (14%) of the initial *n* = 125 patients discontinued participation in the study, leaving *n* = 107 patients for the V1 visit (86%). Between V1 and V2, the number of responding patients fell by *n* = 31 (25%) to *n* = 76 (61%), and *n* = 60 (48%) patients participated in the follow-up visits V3 and V4 (Figure 1).

The study participants fasted for between 3 and 12 days during their hospital stay, with a peak between 7 and 9 days (7.73 ± 1.70 days), cf. Figure 2A. There was a marked reduction in self-reported pain from 6.2 (±1.72) to 3.5 (±1.87) on an NRS ranging from 0 to 10 (corresponding to a drop of −2.7 points or −45%, T = 10.8, *p* < 0.001; cf. Figure 3A and Table 2). This reduction corresponds to a clinically significant improvement, exceeding the threshold for an improvement that feels “much better” for the patient. Fasting also resulted in an average weight loss of 3.6 (±1.65) kg between the first day after hospital admission (day 1) and V1 (T = 23.3, *p* ≤ 0.001; cf. Figure 3B and Table 2). We decided against using the weight measure at V0 (admission day), as patients did not arrive at admission in a fasted state, and, thus, participants had to be weighed on the morning of the day after admission (d01) for a valid measurement to be obtained by the ward personnel (cf. Table 2). The hospital stay was also associated with a discrete drop in systolic (−6.2 ± 15.9 mmHg, T = 4.2, *p* < 0.001) and a somewhat smaller reduction in diastolic blood pressure (−3.7 ± 10.6 mmHg, T = 3.7, *p* < 0.001) between V0 and V1 (Figure 3C and Table 2). Serum cholesterol levels sank noticeably from 239.2 ± 44.86 mg/dL to 201.1 ± 48.85 mg/dL (T = 10.2, *p* < 0.001), as did levels of LDL (155.9 ± 38.27 mg/dL to 129.8 ± 46.91 mg/dL, T = 6.29, *p* < 0.001), HDL (58.2 ± 13.80 mg/dL to 49.2 ± 12.61 mg/dL, T = 8.12, *p* < 0.001), and triglycerides (133.7 ± 66.42 mg/dL to 113.1 ± 47.25 mg/dL, T = 2.92, *p* = 0.005).

Pain medication was categorized into non-steroidal anti-inflammatory drugs (NSAID) on the one hand—including ibuprofen, diclofenac, coxibs, paracetamol, and metamizole, among others—and opioids/opiates on the other. Medications targeting neuropathic pain components such as carbamazepin, gabapentin, biologicals, and corticosteroids were listed separately, as were herbal remedies. For the first two categories (NSAIDs and opioids), we applied a self-developed scale to approximate whether the dosage had been raised or reduced. We used a scale consisting of scores of −2 (medication was stopped), −1 (dosage was reduced significantly, including, i.e., stopping rescue medication or reducing it from daily intake to rescue medication only), 0 (no noteworthy change in dosage), +1 (new medication or a 1.5 to 2.5-fold rise in dosage), and +2 (at least a 3-fold increase in medication) and rated both categories separately (cf. Figure 2B–D). Regarding the medication used for neuropathic pain components, in *n* = 3 patients, medication remained stable, while *n* = 1 patient had their dosage reduced during their hospital stay. Biologicals or immunosuppressants were taken by *n* = 2 patients, and the dosage remained stable during the inpatient stay. N = 5 patients were taking corticosteroids at the beginning of fasting, and two reduced their medication notably during the stay, while one patient received corticosteroids as a new treatment during their hospital stay. Herbal remedies were only taken by a minority of the patients at the time of admission, and were prescribed to a greater number of patients during their hospital stay (Table 3). Concerning herbal medication, in most cases, a tincture containing Faxinus excelsior, Populus tremula, and Solidago virgaurea (Phytodolor^®^) was used. Other herbal remedies prescribed were Curcumin, Boswellia serrata, and Harpagophytum procumbens. CBD oil (10%) was used in 10 cases, and 1 patient was prescribed Dronabinol. The transition from the previously used medication to a herbal remedy was made by shared decision making between the physician and the patient, mainly based on self-reported pain during visits.

Self-reported adverse effects of fasting were reported mainly during the first few days, and included mild headaches in approximately 50% of cases, feelings of hunger and reduced mood in approximately 20%, and gastrointestinal complaints and stronger headaches/migraines in a further 10% of the patients. However, counterchecking these complaints in the medical records revealed that, except for mild headaches and nausea, most of these complaints were not reported to medical staff or nurses (see Figure 3d).

The pain reduction measured by the NRS shown above was mirrored by the results of the WOMAC Index. The overview in Figure 4A–D shows that pain, joint stiffness, and constraints of physical function were, both in each WOMAC subscale and globally, significantly reduced by approximately 40% during the hospital stay. In detail, the global WOMAC global score was reduced by 14.9 points (on the 0–96 scale), thus exceeding the minimum clinically important difference (MCID) of 4. The finer depiction in Figure 3E–H indicates that this was true for all OA patients taken together as a group (black bars), as well as for those with knee OA (dark green bars) vs. hip OA (light green bars) separately. The differences between the two subgroups were not significant.

Figure 4A–D additionally show long-term effects. As expected, the strong effect faded, as was observed in the follow-up visits after 3, 6, and 12 months (V2, V3, and V4, respectively). However, even after 12 months (V4), the values for stiffness were still reduced by 15% in comparison to baseline values, while the effects were even better on the WOMAC Global Index score as well as on the subscales for pain and physical function, with a remaining significant reduction of 25% compared to baseline values (for details see Table 4). The reductions between baseline and V2 for pain, and between baseline and all visits for the other three subscales, represent moderate-sized effects (Cohen’s d ≥ 0.50).

Similar effects were observed for the quality of life measured by the WHO-5 questionnaire (Figure 4E). The score increased from 11.1 (±4.74) at baseline to 15.4 (±4.75; T = 9.3, *p* < 0.001), and remained at an elevated level (14.3 ± 4.61) until V4, again with moderate effect sizes of d ≥ 0.5.

The increase in the MAAS questionnaire score measuring mindfulness (3.9 ± 0.87 to 4.1 ± 0.77, T = 4.1, *p* < 0.001) had a smaller effect size (d = 0.29) and lasted until V4 (Figure 4F).

Notable reductions were also registered for anxiety (10.3 ± 3.73 to 8.2 ± 3.69, T = 7.28, *p* < 0.001; cf. Figure 4G) and depression (10.89 ± 3.74 to 8.5 ± 3.32, T = 7.9, *p* < 0.001, cf. Figure 4h), as measured by the HADS depression and anxiety questionnaire. Both effect sizes were moderate, between V0–V1. The effects remained on a similar level (d > 0.5) until after the first follow-up visit (V2).

In a final test, we checked whether the positive long-term results could be due to a selective loss to follow-up of those who did not gain strongly from the interventions. However, identifying and tracking those who gained most, moderately, and least revealed that losses to follow-up occurred to a larger extent with patients that had gained either the most or the least during the stay, and to a smaller extent in the central group. In conclusion, the long-term results seem not to stem from a selective loss of those who had gained less from the intervention.

In B–D: (−2) = medication was stopped; (−1) = dosage was substantially reduced; (−0.5) = change from daily intake of herbal remedies to rescue medication; (0) = no noteworthy change in dosage; (+0.5) = slight increase in dosage or new herbal remedies as rescue medication; (+1) = new medication or a 1.5 to 2.5-fold rise in dosage; (+1.5) = new daily herbal remedy plus an herbal remedy as rescue medication; and (+2) = at least a 3-fold increase in medication, or two new daily herbal remedies.

Overall, black bars indicate means and SDs for the whole group, while green and light green bars show results for the patients suffering mainly from knee and hip OA, respectively. In (d), black bars show percentages of the side effects mentioned in the questionnaires, while gray bars indicate percentages of the complaints expressed towards the ward team or physicians, as documented in patient files. V0, V1, V2, V3, and V4 = visits at the beginning and the end of the fasting stay and after 3, 6, and 12 months. d00 indicates the day of admission (noon) and d01 the day after admission, on which the measurements were made by a standardized method in the morning.

## 4. Discussion

This explorative, single-arm, prospective, single-center, open-label, observational study on prolonged modified fasting with a max. intake of 600 kcal daily and an average duration of 8 days was embedded in a complex therapeutic inpatient CIM/TEM intervention. It resulted in improvements in several parameters relevant to osteoarthritis of the knee and hip. Relevant improvements were observed for patient-reported outcomes on pain, functionality, and quality of life, as well as for clinical, anthropometric, and laboratory parameters.

In osteoarthritis, pain is often one of the leading symptoms that makes up for much of the disease burden and need for medication. Our data show a decrease in subjective and objective measures of pain with clinical significance. In the WOMAC Index subscale for pain, we observed a drop in experienced pain from hospital admission to discharge, and the effect seemed to last until one year post-intervention. At the same time, data derived from the patient files demonstrated a pronounced reduction of reported pain on a numerical rating scale, exceeding a clinical rating of feeling “much better”. Dosage of pain medication either remained stable or was reduced in favor of the prescription of milder herbal remedies.

Functionality is another aspect that is important to OA patients. Impaired functionality not only potentially lowers quality of life in itself, but also often contributes to reduced mobility and exercise and, thus, may increase cardiovascular and metabolic disease risk. Our results showed a marked decrease in joint stiffness and physical impairment in the WOMAC Index subscales of stiffness and physical function for the entire year post-intervention, with the effects lessening slightly over the course of time.

We suggest three main pathomechanisms for symptom reduction in OA through fasting. One is the anti-inflammatory effect of fasting, which has been demonstrated in animal and human studies [5,13,34,54]. This is probably mediated by cellular stress response mechanisms such as autophagy, mitophagy, and sirtuine activation, as well as systemic hormonal and metabolic responses to nutrient deprivation [5,55]. The pathogenesis and symptomatology of OA seem to be partly mediated by low-grade inflammatory processes [56]. Additionally, during fasting, patients are educated on healthy nutrition so that some may shift to a more plant-based diet following the fasting period. Healthy plant-based diets have been shown to possess anti-inflammatory properties [4]. Following a Mediterranean diet, which is also known to reduce systemic inflammation, has likewise shown symptom relief in OA patients [57]. Secondly, fasting contributes to weight loss, as we also saw in our sample [5]. This may lessen the mechanical load on the joints of the lower extremities, possibly contributing to pain reduction [58]. Furthermore, lipid metabolism is improved by fasting [59]. Even in patients with normal BMI, metabolic factors have been shown to be associated with disease severity [60]. It seems that, apart from the aforementioned systemic low-grade inflammation associated with higher levels of visceral fat, cholesterol metabolism and adipokines play a pivotal role in disease progression as they activate diverse cartilage-degrading mechanisms [58,60,61,62]. Thirdly, fasting has been shown to possess antioxidative capacities that can also make up for some of the effects, as has been described for other nutritional therapeutic approaches to OA [4,5]. Oxidative processes have, both in vitro and in vivo, been found to impact pathogenesis and disease severity in OA [62,63]. To what extent the enhancement of serotonin pathways in the central nervous system and the switch in the microbiota that have been described in prolonged fasting play a role in this context cannot be determined yet [6,35].

Taking the positive findings on pain and functionality together, we expected to see quality of life increase in our data set. Quality of life as measured by WHO-5, as well as depression and anxiety as measured by the HADS questionnaire, showed marked improvements from V0 to V1 that were sustained for up to one year. This finding is in line with those from other studies that have shown that fasting positively affects mood and quality of life [30,64,65].

The measured metabolic parameters included anthropometric and laboratory parameters, such as weight and blood lipids. All decreased notably in the eight average days of fasting. This change was not only positive because of its swiftness and its aforementioned effects on OA symptomatology, but may also merit attention because OA patients tend to carry a higher cardiovascular risk [3].

There were several limitations to this study. Firstly, there was no control group, a fact that significantly restricts the interpretability of the results. It was not possible to randomize inpatients to different groups, as hospitalized patients need and expect the optimal individual therapy possible; therefore, we chose an observational study design. All inpatients who were not prescribed a fasting intervention were assigned so due to contraindications, such as being underweight or having eating disorders, and would not have served as a sensible control group. As dietary interventions do not lend themselves to blinding per se—neither for patients nor for medical personnel—this is one more methodological limitation which we were not able to avoid. Since the hospital program allows for individualized complex CIM/TEM therapies, it is difficult, if not impossible, to differentiate between the effects of fasting and those of the other interventions. However, in its traditional form, prolonged therapeutic fasting is regularly accompanied by some exercise and mind–body interventions. This is practiced to overcome hunger or other adverse effects, both to support fasting adherence and to frame the exceptional experience of renouncing solid food for several days or even weeks [16]. Additionally, fasting length was determined by a shared decision between the therapeutic team and the patient, and was based on traditional medicine rationales such as individual constitution as defined by the paradigms of TEM. Obviously, this reduced the study’s reproducibility. To make treatments more comparable in the future, it would be necessary to define the fasting length or at least to document the criteria for decision making in more detail. In addition, the dietary changes of patients during the follow-ups were not recorded, which made it impossible to differentiate the effects of fasting per se from any dietary changes that shaped their eating habits after the hospital stay. Furthermore, we did not differentiate OA from the commonly used Kellgren–Lawrence scale or other classifications. Finally, the limitations of questionnaire follow-ups always include fewer responses over the course of time. However, in this study, the response rates were relatively good, and the answers did not only come from patients who profited most from the hospital stay.

Considering all of these limitations, our results can only serve as preliminary data showing the feasibility, safety, and potential effects of a traditional therapeutic fasting approach in knee and hip OA patients. The safety and feasibility of fasting therapies have been shown for other indications, such as rheumatoid arthritis, fibromyalgia syndrome, diverse cardiometabolic conditions, and type 2 diabetes mellitus [7,30,38,43]. Effects on disease symptomatology have also been shown previously in a small cohort of 30 OA patients, which included 20 patients with OA of the lower limbs [11].

In summary, fasting may hold promising therapeutic potential for both the immediate and the secondary disease burden of knee and hip OA. Future studies should, on the one hand, provide control groups and more rigorously controlled conditions. On the other hand, the pragmatic exploration of therapeutic fasting or fasting-mimicking diets in outpatient settings under the patients’ real-life conditions seems equally warranted. From a health economy perspective, it would be interesting to investigate cost-effectiveness, considering savings related to medication and its side-effects, necessity for other therapies, as well as sick leave. If a short-term dietary intervention such as a 5–10 day fast could influence joint health in a clinically meaningful and sustainable way, it seems worthwhile for this method to be explored further.

## 5. Conclusions

Prolonged modified fasting could potentially support patients with OA of the knee or hip as part of an integrative multimodal approach for this common chronic condition.

## Figures and Tables

**Figure 1 nutrients-15-02695-f001:**
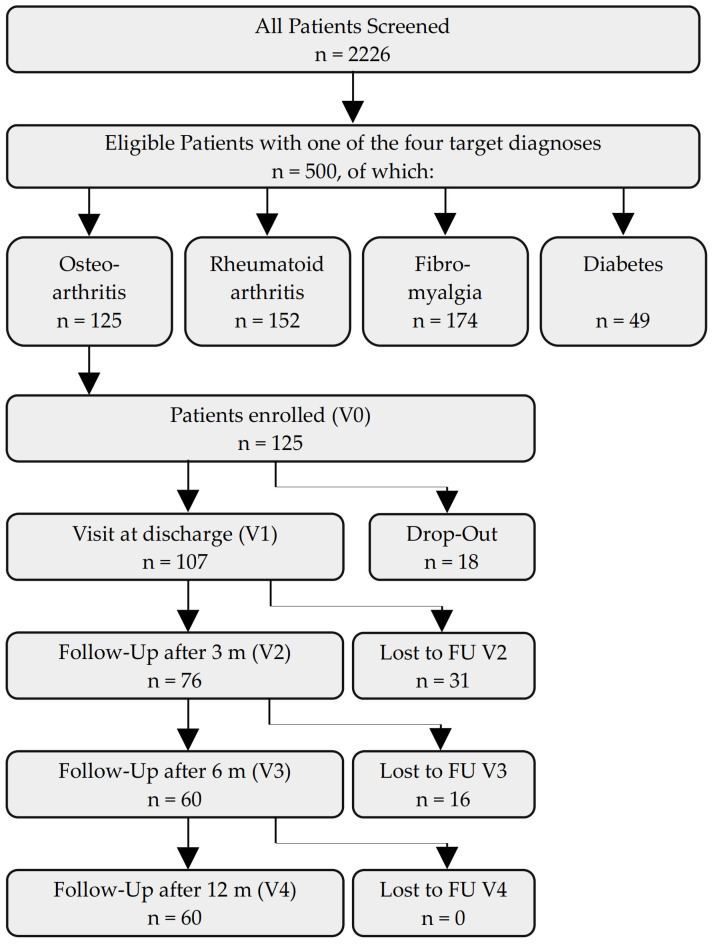
Flowchart of participants. Legend: FU = follow-up, m = months, V = visit.

**Figure 2 nutrients-15-02695-f002:**
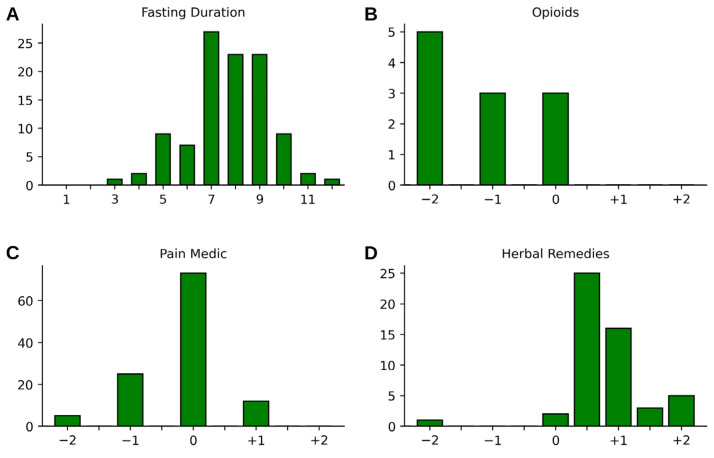
(**A**) Histograms of the fasting duration (in days) and changes to the intake of pain medications during the inpatient hospital stay: (**B**) opioids, (**C**) other pain medication, and (**D**) herbal remedies.

**Figure 3 nutrients-15-02695-f003:**
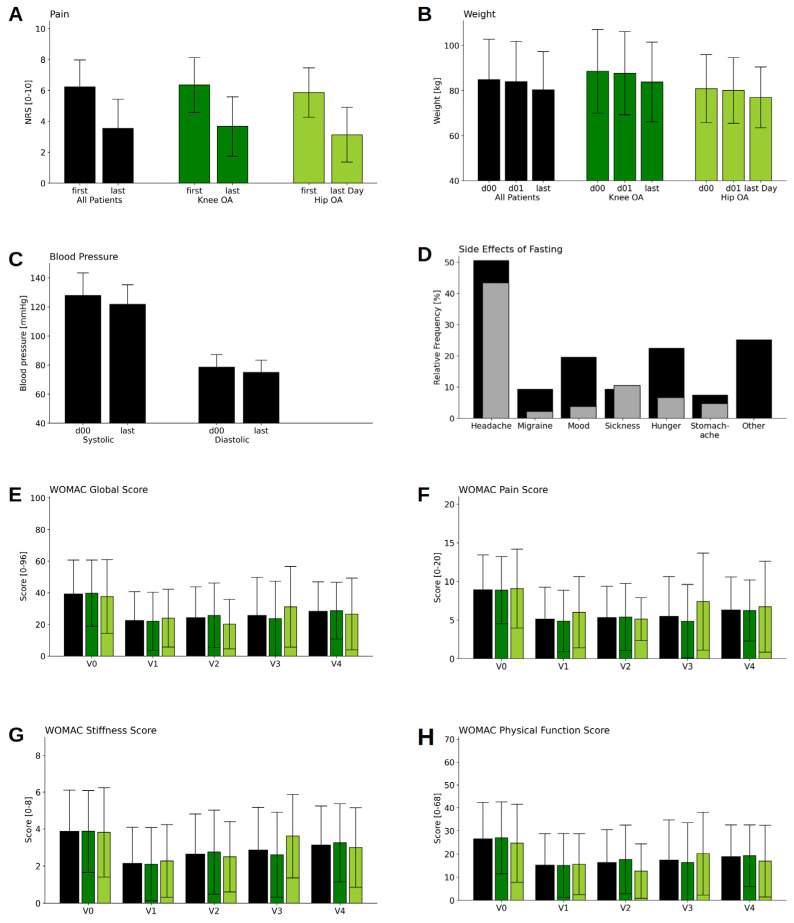
Results for pain and physiological data (**A**–**C**), side effects (**D**), and the WOMAC Index (on the traditional 0–96 point scale) (**E**–**H**).

**Figure 4 nutrients-15-02695-f004:**
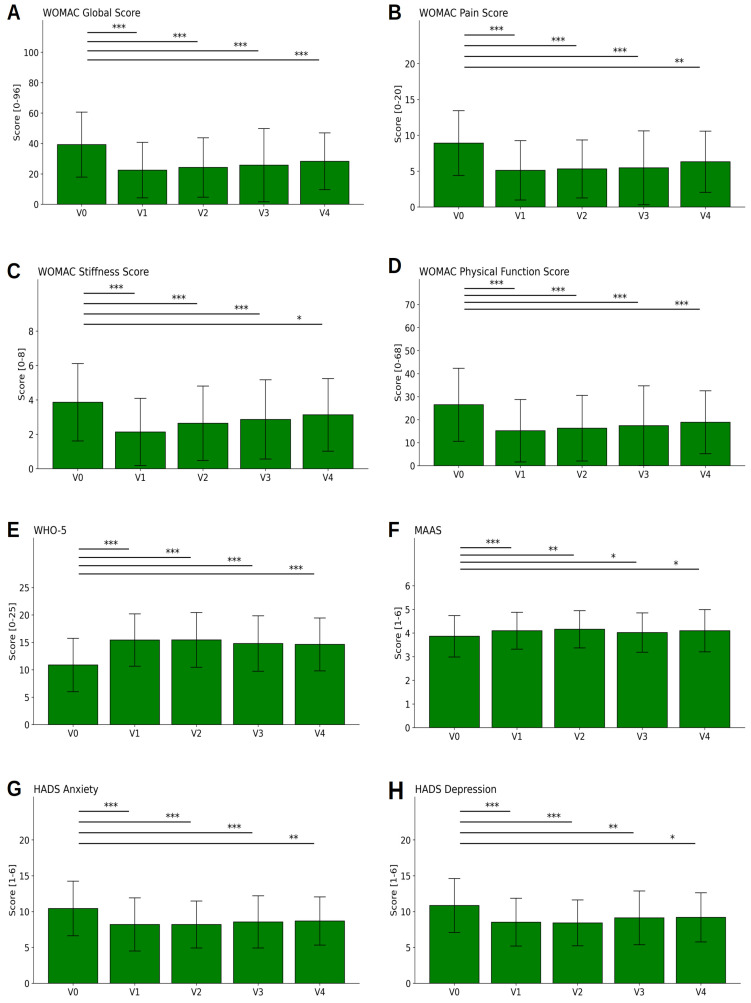
Questionnaire results. Results from the validated WOMAC Index (on the traditional 0–96 point scale) (**A**–**D**), WHO-5 (**E**), MAAS (**F**), and HADS (**G**,**H**) across the five visits (V0, V1, V2, V3, V4), i.e., at the beginning and the end of the fasting stay and after 3, 6, and 12 months. Higher values for WHO-5 (**E**) and MAAS (**F**) and lower values for all other scores indicate better results in terms of the patients’ health. Asterisks *, **, *** indicate tests with *p* < 0.05, *p* < 0.01, and *p* < 0.001, respectively.

**Table 1 nutrients-15-02695-t001:** Baseline characteristics.

Parameter	Value	All Patients	Knee OA	Hip OA
Total		125 (100.0%)	97 (100.0%)	28 (100.0%)
Sex	Female	107 (85.6%)	81 (83.5%)	26 (92.9%)
	Male	18 (14.4%)	16 (16.5%)	2 (7.1%)
Age group (years)	18–35	2 (1.6%)	2 (2.1%)	0 (0.0%)
	36–50	8 (6.4%)	5 (5.2%)	3 (10.7%)
	51–65	77 (61.6%)	64 (66.0%)	13 (46.4%)
	66–80	38 (30.4%)	26 (26.8%)	12 (42.9%)
Marital status	single	16 (12.8%)	14 (14.4%)	2 (7.1%)
	married	69 (55.2%)	51 (52.6%)	18 (64.3%)
	separated or divorced	29 (23.2%)	23 (23.7%)	6 (21.5%)
	widowed	9 (7.2%)	7 (7.2%)	2 (7.1%)
	other	2 (1.6%)	2 (2.1%)	0 (0.0%)
Household	single	43 (34.4%)	36 (37.1%)	7 (25.0%)
	with partner	61 (48.8%)	43 (44.3%)	18 (64.3%)
	single with children	4 (3.2%)	4 (4.1%)	0 (0.0%)
	with partner and children	15 (12.0%)	12 (12.4%)	3 (10.7%)
	other	2 (1.6%)	2 (1.2%)	0 (0.0%)
Highest educational level	primary schooling	8 (6.4%)	6 (6.2%)	2 (7.1%)
	secondary schooling	34 (27.2%)	26 (26.8%)	8 (28.6%)
	high school	21 (16.8%)	17 (17.5%)	4 (14.3%)
	university degree	56 (44.8%)	43 (44.3%)	13 (46.4%)
	other	6 (4.8%)	5 (5.2%)	1 (3.6%)
Occupation	self-employed	12 (9.6%)	10 (10.3%)	2 (7.1%)
	civil servant	5 (4.0%)	4 (4.1%)	1 (3.6%)
	employed	39 (31.2%)	31 (32.0%)	8 (28.6%)
	worker	2 (1.6%)	1 (1.0%)	1 (3.6%)
	homemaker	3 (2.4%)	3 (3.1%)	0 (0.0%)
	unemployed	5 (4.0%)	5 (5.2%)	0 (0.0%)
	retired	43 (34.4%)	30 (30.9%)	13 (46.4%)
	permanently disabled	12 (9.6%)	10 (10.3%)	2 (7.1%)
	other	4 (3.2%)	3 (3.1%)	1 (3.6%)
Annual gross salary	<20.000 Euros	49 (39.2%)	40 (41.2%)	9 (32.1%)
	20–40.000 Euro	42 (33.6%)	33 (34.0%)	9 (32.1%)
	40–60.000 Euro	21 (16.8%)	14 (14.4%)	7 (25.0%)
	60–80.000 Euro	11 (8.8%)	8 (8.2%)	3 (10.7%)
	>80.000 Euro	2 (1.6%)	2 (2.1%)	0 (0.0%)
Subjective physical health status	not impaired	1 (0.8%)	1 (1.0%)	0 (0.0%)
	mildly impaired	16 (12.8%)	11 (11.3%)	5 (17.9%)
	impaired	76 (60.8%)	59 (60.8%)	17 (60.7%)
	strongly impaired	32 (25.6%)	26 (26.8%)	6 (21.4%)
Subjective psychological health status	not impaired	21 (16.8%)	15 (15.5%)	6 (21.4%)
	mildly impaired	49 (39.2%)	36 (37.1%)	13 (46.4%)
	impaired	38 (30.4%)	32 (33.0%)	6 (21.4%)
	strongly impaired	17 (13.6%)	14 (14.4%)	3 (10.7%)
Psychotherapy	none so far	49 (39.2%)	37 (38.1%)	12 (42.9%)
	ealier	57 (45.6%)	45 (46.4%)	12 (42.9%)
	currently	19 (15.2%)	15 (15.5%)	4 (14.3%)
Integrative medicine	familiar with concept	86 (68.8%)	65 (67.0%)	21 (75.0%)
Stay at this clinic	first	83 (66.4%)	67 (69.1%)	16 (57.1%)
	second	23 (18.4%)	16 (16.5%)	7 (25.0%)
	third	12 (9.6%)	8 (8.2%)	4 (14.3%)
	fourth	7 (5.6%)	6 (6.2%)	1 (3.6%)
Fasting experience	never	56 (44.8%)	44 (45.4%)	12 (42.9%)
	once	17 (13.6%)	12 (12.4%)	5 (17.9%)
	twice	18 (14.4%)	14 (14.4%)	4 (14.3%)
	3 times	9 (7.2%)	7 (7.2%)	2 (7.1%)
	4 times	5 (4.0%)	5 (5.2%)	0 (0.0%)
	5 times and more	20 (16.0%)	15 (15.5%)	5 (17.9%)
Medication at admission	Opioids	11 (8.8%)	8 (8.2%)	3 (10.7%)
	Pain Medication	115 (92.0%)	88 (90.7%)	27 (96.4%)
	Herbal Remedies	52 (41.6%)	40 (41.2%)	12 (42.9%)
Subjective impairment by OA	NRS [0–10]: M ± SD	6.1 (±1.6)	6.2 (±1.6)	5.9 (±1.5)
Anticipation of efficacy	NRS [0–10]: M ± SD	6.4 (±2.0)	6.4 (±1.9)	6.4 (±2.5)

Legend: M = mean, SD = standard deviation, NRS = numerical rating scale, OA = osteoarthritis.

**Table 2 nutrients-15-02695-t002:** Results of metabolic and physiological parameters.

					Difference between V1 and V0 *				
Parameter	Visit	M	SD	*n*	M	SD	T	*p*	d
Cholesterol (mg/dL)	V0	239.2	44.86	84					
	V1	201.1	48.85	70	−38.4	31.20	10.23	<0.001	0.80
LDL (mg/dL)	V0	155.9	38.27	81					
	V1	129.8	46.91	63	−24.5	30.72	6.29	<0.001	0.56
HDL (mg/dL)	V0	58.2	13.80	81					
	V1	49.2	12.61	62	−8.0	7.74	8.12	<0.001	0.59
Triglycerides (mg/dL)	V0	133.7	66.42	84					
	V1	113.1	47.25	66	−24.5	67.43	2.92	0.005	0.42
NRSPain (scale 0–10)	V0	6.2	1.72	90					
	V1	3.5	1.87	64	−2.7	1.98	10.8	<0.001	1.48
Weight (kg)	V0	84.8	17.9	115					
	D 01	83.9	17.74	115					
	V1	80.3	16.88	115	−3.6	1.65	23.29	<0.001	0.21
Systolic BP (mmHg)	V0	128.0	15.34	115					
	V1	121.8	13.42	115	−6.2	15.93	4.15	<0.001	0.43
Diastolic BP (mmHg)	V0	78.6	8.61	115					
	V1	74.9	8.4	115	−3.7	10.55	3.70	<0.001	0.43

* between V1 and the day after hospital admission (day 01) in the case of weight. Legend: Differences and statistics were calculated only for complete cases regarding each individual parameter. Since patients were admitted at noon, weight was assessed on the morning after admission (day 1) and V1. M = mean, SD = standard deviation, *n* = number of participants, *T* = test statistic, *p* = *p*-value of the paired *t*-test, d = effect size (Cohen’s d), NRS = numerical rating scale, LDL = low-density lipoprotein, HDL = high-density lipoprotein, BP = blood pressure. Units: mg = milligram, dL = deciliter (0.1 L), kg = kilogram, mmHG = millimeter/mercury.

**Table 3 nutrients-15-02695-t003:** Changes in medication during inpatient stay.

Change	−2	−1	−0.5	0	+0.5	+1	+1.5	+2
Opioids	5	3	0	3	0	0	0	0
Pain Medication	5	25	0	73	0	12	0	0
Herbal Remedies	1	0	0	2	25	16	3	5

Legend: (−2) = medication was stopped; (−1) = dosage was significantly reduced; (−0.5) = change from daily intake of herbal remedies to rescue medication; (0) = no noteworthy change in dosage; (+0.5) = slight increase in dosage or new herbal remedies as rescue medication; (+1) = new medication or a 1.5 to 2.5-fold rise in dosage; (+1.5) = new daily herbal remedy plus an herbal remedy as rescue medication; and (+2) = at least 3-fold increase in medication, or 2 new daily herbal remedies.

**Table 4 nutrients-15-02695-t004:** Questionnaire results.

					Difference to V0				
Parameter	Visit	M	SD	*n*	M	SD	*T*	*p*	d
WHO 5	V0	11	4.74	107					
	V1	15.4	4.75	107	4.5	4.94	9.29	<0.001	0.94
	V2	15.3	4.96	68	3.6	4.27	7	<0.001	0.74
	V3	14.9	5.04	56	3.4	4.54	5.57	<0.001	0.67
	V4	14.3	4.61	53	2.7	5.05	3.8	<0.001	0.56
MAAS	V0	3.9	0.87	107					
	V1	4.1	0.77	107	0.2	0.61	4.07	0.001	0.29
	V2	4.2	0.79	68	0.2	0.6	3.28	0.002	0.28
	V3	4	0.81	56	0.1	0.57	1.87	0.067	0.18
	V4	4	0.91	53	0.2	0.71	1.98	0.053	0.21
HADS Depression	V0	10.9	3.74	107					
	V1	8.5	3.32	107	−2.3	3.01	7.94	<0.001	0.65
	V2	8.5	3.17	68	−2	3.13	5.35	<0.001	0.59
	V3	9.1	3.64	56	−1.5	3.3	3.29	0.002	0.37
	V4	9.5	3.37	53	−0.9	2.76	2.32	0.024	0.24
HADS Anxiety	V0	10.3	3.73	107					
	V1	8.2	3.69	107	−2.1	2.91	7.28	<0.001	0.55
	V2	8.3	3.18	68	−1.7	2.4	5.91	<0.001	0.52
	V3	8.6	3.6	56	−1.7	2.84	4.38	<0.001	0.45
	V4	9	3.36	53	−1.1	2.96	2.76	0.008	0.32
WOMAC Global Score (0–96)	V0	37.7	19.33	107					
	V1	22.5	18.16	101	−14.9	13.37	11.18	<0.001	0.79
	V2	23.3	17.25	68	−12.9	13.84	7.65	<0.001	0.71
	V3	24.6	22.98	56	−11.4	20.17	4.19	<0.001	0.53
	V4	28	17.98	53	−10	18.5	3.89	<0.001	0.55
WOMAC Pain Score (0–20)	V0	8.5	3.98	107					
	V1	5.1	4.12	101	−3.4	3.61	9.29	<0.001	0.82
	V2	5.1	3.64	68	−3.1	3.67	6.98	<0.001	0.81
	V3	5.2	5.02	56	−2.7	4.42	4.58	<0.001	0.59
	V4	6.3	4.24	53	−2	4.57	3.1	0.003	0.47
WOMAC Subscale Stiffness Score (0–8)	V0	3.8	2.18	107					
	V1	2.1	1.94	101	−1.6	1.81	8.79	<0.001	0.76
	V2	2.6	2.03	68	−1.2	1.83	5.26	<0.001	0.56
	V3	2.7	2.14	56	−1	1.92	3.73	<0.001	0.46
	V4	3.2	2.11	53	−0.6	2.42	1.91	0.062	0.31
WOMAC Physical Function Score (0–68)	V0	25.4	14.5	107					
	V1	15.2	13.5	101	−10	9.8	10.22	<0.001	0.71
	V2	15.6	12.73	68	−8.6	9.93	7.09	<0.001	0.64
	V3	16.6	16.54	56	−7.7	14.96	3.81	<0.001	0.49
	V4	18.5	13.08	53	−7.3	13.08	4.05	<0.002	0.55

Legend: On the left side, descriptive statistics are given for each visit separately, while on the right side, the differences between the respective visit and the baseline visit (V0) are presented. Differences and statistics were calculated only for the complete cases for individual parameters and visits. M = mean, SD = standard deviation, *n* = number of participants, *T* = test statistic, *p* = *p*-value of the paired *t*-test, d = effect size (Cohen’s d).

## Data Availability

Data can be obtained from the corresponding author upon justified request.

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
