# Peer review of "Effects of Prolonged Fasting during Inpatient Multimodal Treatment on Pain and Functional Parameters in Knee and Hip Osteoarthritis: A Prospective Exploratory Observational Study"

_nutrients, 2023, doi:10.3390/nu15122695_

Round 1
Reviewer 1 Report
I consider this article very relevant in the theme “Effects of prolonged fasting during multimodal hospitalization treatment on pain and functional parameters in osteoarthritis of the knee and hip, being innovative because scientific evidence is still scarce on this topic and the discussion is still not very consensual , so this article is mainly a contribution to increasing knowledge and discussion on this subject. However, from my point of view, there must be some care in prolonged fasting, namely unbalanced diets; diets with VCT below Basal Metabolism and also deficiencies of macro and micronutrients. We know that weight reduction in people who are overweight or obese always benefit from losing weight: reduction in blood pressure; reduction of the lipid record; reduced insulin resistance; improvement in cardiovascular risk and improvement in joint pain, but this weight loss cannot be at the expense of nutritional deficiency, we also know the rate of inflammatory diseases is growing in modern society. But we also know that caloric restriction can shape the composition of the microbiota and decrease the expression of inflammatory factors along the gastrointestinal tract. I would also have liked to have seen the assessment of Nutritional Risk included in your study.
I therefore make some suggestions that I believe may be a contribution to clarify and enrich the investigation:
Material and methods
Item 118 …..individual and multimodal treatment, including a dietary regimen-Which?
Item 122 ………validated questionnaires… Which ones?
Item 125……..Laboratory tests included in standard ward procedures.. Which ones?
Item 127………side effects of fasting…..Which ones?
Item 146………caloric reduction of approximately 1200 kcal. …The same for all patients?
Item 147 …..which is induced by taking laxatives, such as Glau-147 salt, or an enema…..Cleaning the bowel for what purpose? To reduce the initial weight?
Item 150…….depending on the patient's individual constitution….. What constitution? Body composition?
Item 151……only natural juices, unsweetened teas consumed ….. Teas or infusions? Teas have theine.
Item 154…..for a diet that mimics fasting…..What is the composition of this diet?
Item 165……nutrition counseling…….Performed by whom? by a Nutritionist?
Item 169 ……depending on the duration of the fast, solid foods……..Which ones?
Item 172…. usually a plant-based normocaloric diet…… what percentage of macronutrients, it has to be balanced!
Item 181…..or sarcopenia, acute psychosis, severe psychiatric pathologies and severe metabolic conditions such as kidney or liver failure were excluded… Not diabetes! What about hypoglycemia?
Results
Item 249 ……..under medical fasting…..It is not noticeable!
Item 278……HDL 58.2 ± 13.80 mg/dL to 49.2 ± 12.61 mg/dL…… HDL high-density lipoprotein lowered which was not a good result should have increased.
Discussion
Item 410….prescription of milder herbal remedies……Which ones?
Item 425……Healthy plant-based diets….. Rich in antioxidants?
Author Response
I consider this article very relevant in the theme “Effects of prolonged fasting during multimodal hospitalization treatment on pain and functional parameters in osteoarthritis of the knee and hip, being innovative because scientific evidence is still scarce on this topic and the discussion is still not very consensual, so this article is mainly a contribution to increasing knowledge and discussion on this subject. However, from my point of view, there must be some care in prolonged fasting, namely unbalanced diets; diets with VCT below Basal Metabolism and also deficiencies of macro and micronutrients. We know that weight reduction in people who are overweight or obese always benefit from losing weight: reduction in blood pressure; reduction of the lipid record; reduced insulin resistance; improvement in cardiovascular risk and improvement in joint pain, but this weight loss cannot be at the expense of nutritional deficiency, we also know the rate of inflammatory diseases is growing in modern society. But we also know that caloric restriction can shape the composition of the microbiota and decrease the expression of inflammatory factors along the gastrointestinal tract. I would also have liked to have seen the assessment of Nutritional Risk included in your study.
Thank you for these comments. In our case, fasting is a therapeutic intervention. It is carried out only based on a sound medical indication after excluding contraindications (s. Methods 2.4) and with appropriate support. Formally, fasting is a short period of time with nutritional scarcity, which in this case is intentional. This is why an assessment of nutritional risk did not seem meaningful to us in this case.
I therefore make some suggestions that I believe may be a contribution to clarify and enrich the investigation:
Material and methods
Item 118 …..individual and multimodal treatment, including a dietary regimen-Which?
There are different dietary regimens available at the inpatient department of IMNT. As mentioned in lines 120 and under 2.4, “all patients who received an inpatient treatment that included therapeutic fasting” were considered eligible for this study. The other dietary regimens do not play any role for the purpose of this study and their descriptions were therefore omitted.
Item 122 ………validated questionnaires… Which ones?
Please refer to „2.5 Variables” for a list of all questionnaires used.
Item 125……..Laboratory tests included in standard ward procedures.. Which ones?
For all laboratory parameters included, please refer to lines 131-136
Item 127………side effects of fasting…..Which ones?
The question which side effects were mentioned by the patients was an open one, without predefined categories. For results on which side effects were noted by patients and/or personnel please refer to Figure 3 and the adjacent text.
Item 146………caloric reduction of approximately 1200 kcal. …The same for all patients?
Yes. If individual options exist, we define that in the text. This is not the case here.
Item 147 …..which is induced by taking laxatives, such as Glau-147 salt, or an enema…..Cleaning the bowel for what purpose? To reduce the initial weight?
This type of bowel cleansing is a traditional intitiation of the fast in Germany. There is no scientific rationale behind it. Patients often report that fasting is easier for them when they use bowel cleansing, but this is only an empirical value. This is why we have changed this procedure lately for it not to be obligatory anymore. During the study though, patients were strongly advised to follow this procedure.
Item 150…….depending on the patient's individual constitution….. What constitution? Body composition?
Body constitution is defined as a MESH term by the National Library of Medicine (https://www.ncbi.nlm.nih.gov/mesh?Db=mesh&Cmd=DetailsSearch&Term=%22Body+Constitution%22%5BMeSH+Terms%5D) as following: “The physical characteristics of the body, including the mode of performance of functions, the activity of metabolic processes, the manner and degree of reactions to stimuli, and power of resistance to the attack of pathogenic organisms.” All these play a role in determining the length of the fast. For example, a patient with a BMI of 24 kg/m2 might have had a very stressful time before coming to the clinic and be exhausted. He might feel very tired at day 6 of the fast, not being able to do the light exercises accompanying the fasting therapy. Here a physician might decide to stop the fast, in agreement with the patient, because the reactivity of the body to the stimulus of fasting might not be as good as it would have been without the exhaustion. Thus, it is not only body composition that plays a role, but it is the whole constitution that is considered.
Item 151……only natural juices, unsweetened teas consumed ….. Teas or infusions? Teas have theine.
Thank you for this comment, we have changed the wording to: “unsweetened herbal teas”
Item 154…..for a diet that mimics fasting…..What is the composition of this diet?
It is a light plant-based diet with reduced protein content. The dishes change every day.
Item 165……nutrition counseling…….Performed by whom? by a Nutritionist?
Yes. To our knowledge, nutritional counseling per definition needs to be carried out by specialized personnel. For a reference, the National Cancer Institute of the US defines nutritional counseling as “A process by which a health professional with special training in nutrition helps people make healthy food choices and form healthy eating habits” (https://www.cancer.gov/publications/dictionaries/cancer-terms/def/nutritional-counseling). Do you think the text would profit from a further clarification?
Item 169 ……depending on the duration of the fast, solid foods……..Which ones?
The reintroduction of solid foods entails complex carbohydrates, easily digestible plant-based protein, and small amounts of plant oils.
Item 172…. usually a plant-based normocaloric diet…… what percentage of macronutrients, it has to be balanced!
Our nutritionists follow international and national guidelines in their counselling. We have no separate concept for our clinic.
Item 181…..or sarcopenia, acute psychosis, severe psychiatric pathologies and severe metabolic conditions such as kidney or liver failure were excluded… Not diabetes! What about hypoglycemia?
Diabetes mellitus type 2 is an indication for fasting. Hypoglycaemia can only occur in patients with insulin or certain oral antidiabetic medication, both of which are paused during fasting (or reduced under strict glucose monitoring). Diabetes mellitus type 2 per se does not engender the risk of hypoglycemia. We seldom have patients with diabetes mellitus type 1, but under strict supervision they are also able to fast, regulating their insulin applications accordingly, in communication with their supervising physician.
Results
Item 249 ……..under medical fasting…..It is not noticeable!
Could you please clarify the question? What exactly is not noticeable?
Item 278……HDL 58.2 ± 13.80 mg/dL to 49.2 ± 12.61 mg/dL…… HDL high-density lipoprotein lowered which was not a good result should have increased.
Thank you for this comment. As total cholesterol decreases significantly, this of course affects all lipoprotein fractions. This is physiological and not to be seen as an increase in cardiovascular risk. Also, acute changes in circulating HDL-C levels may not be as relevant for cardiovascular risk as was formerly postulated, as Kosmas et al. point out:
„… more recent evidence from genetic studies and clinical research has come to challenge the long-standing notion that higher HDL-C levels are always beneficial, while lower HDL-C levels are always detrimental. Thus, it becomes apparent that HDL functionality plays a much more important role in atheroprotection than circulating HDL-C levels.“ (Kosmas CE, Martinez I, Sourlas A, Bouza KV, Campos FN, Torres V, Montan PD, Guzman E. High-density lipoprotein (HDL) functionality and its relevance to atherosclerotic cardiovascular disease. Drugs Context. 2018 Mar 28).
Thus, as relevant as HDL levels may be in a case-control setting, their acute changes seem not to be relevant for the assessment of cardiovascular risk. We hope to have answered your question.
Discussion
Item 410….prescription of milder herbal remedies……Which ones?
Thank you for this question. Mostly a tincture containing Faxinus excelsior, Populus tremula and Solidago virgaurea (Phytodolor ®) was used. Other herbal remedies prescribed were Curcumin, Boswellia serrata and Harpagophytum procumbens. 10% CBD-Oil was used in 10 cases, and one patient was prescribed Dronabinol. We have added this information in the results section.
Item 425……Healthy plant-based diets….. Rich in antioxidants?
In whole food plant-based diets antioxidants are one of the reasons for their positive health effects. Other secondary plant compounds and fibers among others, also contribute to them. There can also be unhealthier plant-based diets, using a lot of ultra-processed food components and glucose/fructose, this is why we chose to not use the plain term “plant-based diets” alone.
We hope to have answered your questions and thank you for helping us improve our manuscript!
Reviewer 2 Report
Nutrients – effects of prolonged fasting during inpatient multimodal treatment on pain and functional parameters in knee and hip osteoarthritis: a prospective exploratory observational study
Abstract
Abstract – clarify at what time point(s) these results were noted (at end of inpatient stay? Or at the 3,6,12 month time points following discharge)
Abstract – perhaps at least mention what other interventions were instituted and if these were standardized across all inpatients or if the only thing that patients had in common was the fasting protocol
Introduction
Suggest to combine first two paragraphs on why OA is an issue – currently there is some repetition which could be streamlined (e.g. states in both first sentences (first and second paragraph) that OA is a common joint disorder worldwide)
Then suggest to combine paragraphs 3 and 4 on the role of nutrition in OA symptomatology and what is known regarding impacts of fasting. Further recommend to include here more specific details on what constitutes fasting: simple caloric restriction, intermittent fasting, duration of fasting when observed signs have been noted?
147 – define TEM?
150 – could you provide a bit more specific information on how fasting duration was determined? Were certain parameters of systemic disease considered too severe for patients to undergo prolonged fasting? Was fasting always performed on consecutive days?
266 – at what time point in the therapeutic plan was this reduction in pain noted? And specify below regarding improvement in other bloodwork parameters
297 – can you provide more information on what constituted herbal remedies? Were all herbal remedy formulations/supplementations the same?
309 – as with above – was this 40% reduction at the end of the fasting period?
Figure 1 and methods – not sure that it is necessary to include that other conditions were considered as not relevant to this manuscript but defer to editor (i.e. RA, fibromyalgia, diabetes) – does raise the question though of was there any overlap between groups such as individuals with both OA and diabetes or were they only included if they had one of the four disease processes?
Can you expand further on how the decision to transition to herbal supplement was made? Based on self-reported pain? Or improvement in other parameters?
Can you please add more information on how OA was diagnosed except for pain symptoms? Did patients have baseline standardized imaging etc? – see in line 470 that grading was not performed? Recommend to clarify that earlier and provide some further discussion of why that was not performed or add this information if it can be gleaned retrospectively from records
421 – remove tracked changes
478 – as above, recommend could likely remove mention of other disease processes as they are not the focus of this manuscript but defer to editor’s preferences
Author Response
Nutrients – effects of prolonged fasting during inpatient multimodal treatment on pain and functional parameters in knee and hip osteoarthritis: a prospective exploratory observational study
Abstract
Abstract – clarify at what time point(s) these results were noted (at end of inpatient stay? Or at the 3,6,12 month time points following discharge)
Thank you for the question. Results were noted in the aforementioned time frame, described in the methods section of the abstract: “questionnaires (were answered) at the beginning and end of inpatient treatment, as well as 3, 6 and 12 months after discharge”. The time frame for the subjective pain rating (NRS) was added in the abstract.
Abstract – perhaps at least mention what other interventions were instituted and if these were standardized across all inpatients or if the only thing that patients had in common was the fasting protocol
Thank you for this comment. We have added the information, that fasting was the only common feature of all treatments.
Introduction
Suggest to combine first two paragraphs on why OA is an issue – currently there is some repetition which could be streamlined (e.g. states in both first sentences (first and second paragraph) that OA is a common joint disorder worldwide)
Thank you, the first two paragraphs have been combined.
Then suggest to combine paragraphs 3 and 4 on the role of nutrition in OA symptomatology and what is known regarding impacts of fasting. Further recommend to include here more specific details on what constitutes fasting: simple caloric restriction, intermittent fasting, duration of fasting when observed signs have been noted?
As described, the role of nutrition in OA symptomatology has not been extensively studied, and the role of fasting in OA symptomatology has even less. This is why we chose to first describe what is known about nutrition in this respect and then to relate it to fasting physiology. Fasting, as described in the reference mentioned (Di Francesco et al., A time to fast) is to be seen as a continuum. Effects mentioned in our list include such as have been described for longer fasts (up to 21 days) as well as intermittent fasts, as many signaling pathways are common to any fasting regimen (please also see Di Francesco et al., A time to fast). Of course, there are also differences, as in our experience there seem to be dose-effect relationships, but these have not yet been extensively described and are therefore omitted for the sake of evidence-based arguments, brevity and focus in this manuscript. We hope this answers your query.
147 – define TEM?
Thank you for this very good question. Traditional European Medicine is not defined as well as, for example, Traditional Chinese Medicine (TCM) is. But the concept of „Naturheilkunde/Naturheilverfahren“ is defined and well established in the German speaking countries. There are corresponding medical reference books such as the “Lehrbuch Naturheilverfahren” (Karin Kraft, Rainer Stange Hrsg, 2009, Hippokrates-Verlag), and these concepts are considered in german medical guidelines, refunding policies of statutory health insurance and in medically certified qualifications. There is no adequate English translation, as naturopathy is often non-medical. Nature-based therapies gets close to it, but TCM also is, in a different way, nature-based. Thus, the wording of TEM has been used over the past years, following the lead of TCM and traditional Indian medicine (TIM), to subsume different traditional medical practices in Europe (“Naturheilkunde/Naturheilverfahren”) under this term, suggesting a separate traditional medical system. We hope to have answered the question.
150 – could you provide a bit more specific information on how fasting duration was determined? Were certain parameters of systemic disease considered too severe for patients to undergo prolonged fasting? Was fasting always performed on consecutive days?
Thank you for pointing this missing information out. We have added in line 154 (methods) that fasting in our clinic is always meant in consecutive days. Please find contraindications for fasting unter 2.4 Participants. Fasting duration is mainly determined by the constitution of the patient, including BMI, body composition, the dynamics of weight loss during fasting and vitality, as well as subjective criteria like well-being, feelings of hunger or exhaustion. These criteria belong to the observational data we collected in the real-world setting and certainly limit the reproducibility of the results. We have added this information in the methods section under 2.3 Interventions, as well as in the limitations section of the discussion.
266 – at what time point in the therapeutic plan was this reduction in pain noted? And specify below regarding improvement in other bloodwork parameters
Thank you for pointing this out. We have added the following text to 2.6 Data collection, where also the information on the bloodwork parameters can be found: “Pain on the NRS scale and side-effects of fasting were documented several times during the inpatient stay. For the calculation of pain scores on the NRS we selected the first and last documented score in the patient record, for most patients this being first and last days of their inpatient stay.”
297 – can you provide more information on what constituted herbal remedies? Were all herbal remedy formulations/supplementations the same?
Thank you for this question. Mostly a tincture containing Faxinus excelsior, Populus tremula and Solidago virgaurea (Phytodolor ®) was used. Other herbal remedies prescribed were Curcumin, Boswellia serrata and Harpagophytum procumbens. 10% CBD-Oil was used in 10 cases, and one patient was prescribed Dronabinol. We have added this information in the results section.
309 – as with above – was this 40% reduction at the end of the fasting period?
This, as pointed out above, is for the whole inpatient stay, including the time after the fasting period.
Figure 1 and methods – not sure that it is necessary to include that other conditions were considered as not relevant to this manuscript but defer to editor (i.e. RA, fibromyalgia, diabetes) – does raise the question though of was there any overlap between groups such as individuals with both OA and diabetes or were they only included if they had one of the four disease processes?
Thank you for this comment. We have discussed it with one of the editors. As publications on the other diagnoses will follow, we thought it would be best to be transparent about the whole study population and the similarity of the setting and methods. This is why we would maintain Figure 1 and methods as they are and hope this meets your expectations. Under 2.4 Participants we included the inclusion criteria of OA as main diagnosis. As described in 2.2 Setting, the main diagnosis for the inpatient stay is one, and according to this the patients were selected into the group. Thus, there is no overlap between the four diagnostic groups. We hope this clarifies the question.
Can you expand further on how the decision to transition to herbal supplement was made? Based on self-reported pain? Or improvement in other parameters?
The goal of the treatment of pain in our clinic is twofold. The restriction of pain medication for the prevention of renal failure, gastric ulcers, and coronary heart disease is one aspect. The other is that pain should be tolerable to the patient. The pain relief many patients experience during fasting seems to be a window of opportunity, to switch from the used medication to herbal remedies, that usually have a multi-drug multi-target mechanism of action and thus show less side-effects. The transition is done by shared decision making, mainly based on self-reported pain during visits. We have added this information in the results section.
Can you please add more information on how OA was diagnosed except for pain symptoms? Did patients have baseline standardized imaging etc? – see in line 470 that grading was not performed? Recommend to clarify that earlier and provide some further discussion of why that was not performed or add this information if it can be gleaned retrospectively from records
Imaging is not routinely performed on the IMNT ward because the patients arrive in the ward prediagnosed earlier by specialists such as orthopedic surgeons or rheumatologists. We have added this information to the methods section under 2.2 Settings.
421 – remove tracked changes
Tracked changes have been removed.
478 – as above, recommend could likely remove mention of other disease processes as they are not the focus of this manuscript but defer to editor’s preferences
Thank you for this comment. Here we have mentioned for which diagnoses fasting has shown feasibility and safety, as both factors have often been questioned from colleagues. In communication with one of the editors for the special edition we have decided to keep this reference and hope the manuscript still meets your expectations.
We hope to have answered your questions and thank you for helping us improve our manuscript!

Reviewer 3 Report
Comments to the Author
In my opinion, article requires general improvement. After corrections it may be reconsidered for publication.
1. The full name of the abbreviation RCT, BMI in articles when the abbreviation is first used.
2. In the text, reference numbers should be placed in square brackets [ ], and placed before the punctuation; for example [1], [1–3] or [1,3].
3. Citing literature should be at the end of the sentence.
4. Revise the literature as required by the journal.
Author Response
Comments to the Author
In my opinion, article requires general improvement. After corrections it may be reconsidered for publication.
- The full name of the abbreviation RCT, BMI in articles when the abbreviation is first used.
Thank you, the text has been adapted.
- In the text, reference numbers should be placed in square brackets [ ], and placed before the punctuation; for example [1], [1–3] or [1,3].
Thank you, brackets have been adapted.
- Citing literature should be at the end of the sentence.
Has been adapted where applicable.
- Revise the literature as required by the journal.
The literature has been formatted in the MDPI format.
We hope to have answered your questions and thank you for helping us improve our manuscript!
